# Research Progress and Hotspot Analysis of Residential Carbon Emissions Based on CiteSpace Software

**DOI:** 10.3390/ijerph20031706

**Published:** 2023-01-17

**Authors:** Yi Chen, Yinrong Chen, Kun Chen, Min Liu

**Affiliations:** College of Public Administration, Huazhong Agricultural University, Wuhan 430700, China

**Keywords:** residential carbon emissions, carbon emissions, resident consumption, CiteSpace, literature review

## Abstract

Residential carbon emissions are one of the critical causes of climate problems such as global warming. It is significant to explore the development and evolution trend of residential carbon emissions research for mitigating global climate change. However, there have been no studies that comprehensively review this research field. Based on the research papers on residential carbon emissions included in the Web of Science core database and China National Knowledge Infrastructure database, the CiteSpace bibliometric analysis software was used in this paper to draw the visual knowledge map of residential carbon emissions research and reveal its research status, research hotspots, and development trend. We found that residential carbon emissions research has gone through the stage of “emergence–initiation–rapid development”, and the research in the United States and the United Kingdom has played a fundamental role in developing this research field. Research hotspots mainly focus on analyzing energy demand, quantitative measurement, and impact mechanisms of residents’ direct and indirect carbon emissions and low-carbon consumption willingness. The focus of research has gradually shifted from qualitative analysis based on relevant policies to the analysis of quantitative spatiotemporal measurements and drive mechanisms of direct and indirect carbon emissions from residential buildings, transportation, and tourism based on mathematical models and geographic information system technologies. Modern intelligent means such as remote sensing technology and artificial intelligence technology can improve the dynamics and accuracy of this research, but there are few related types of research at present. Based on these research status and trends, we proposed that the future research direction of residential carbon emissions should focus more on spatial analysis and trend prediction based on intelligent methods under a low-carbon background.

## 1. Introduction

Since the Industrial Revolution, human activities have consumed a large amount of fossil fuels, which has led to a sharp increase in emissions of heat-absorbing greenhouse gases such as carbon dioxide which, while satisfying the required products and services, caused a large amount of long-wave heat radiation to be absorbed by the atmosphere, thus causing the greenhouse effect. The largest contributor to the greenhouse effect is carbon dioxide, at about 25%, and its concentration tends to increase gradually. Especially since the 18th century, the concentration of carbon dioxide produced by human activities has increased by 31 percent [1]. As a result, the phenomenon of global temperature warming is becoming more and more apparent; the rate of glacier melting, sea level rise, and forest degradation around the world is increasing; extreme weather events such as typhoons and heavy rains are frequent; and the resistance and quality of the natural ecological environment are declining. It is also a huge threat to public health and human well-being. The world is facing severe climate and environmental problems [2,3].

In order to meet this challenge and reduce the impact of the greenhouse effect, countries around the world began to seek new cooperation, and low carbon emission reduction has become one of the focus issues of global policymakers and academic researchers. In December 1997, the third Conference of the Parties to the United Nations Framework Convention on Climate Change was held in Kyoto, Japan, and the Kyoto Protocol was adopted to limit the greenhouse gas emissions of developed countries in order to curb global warming. The agreement committed all developed countries to reduce carbon dioxide emissions by 5.2 percent from 1990 levels, and the need for voluntary cooperation among countries, with financial, technological, and capacity-building support from developed countries to developing countries being emphasized. In December 2015, many countries signed the Paris Agreement at the United Nations Climate Change Conference and pledged to take action to achieve the goal of “keeping the global average temperature rise below 1.5 °C compared with the industrial period” [4]. According to the Sixth Assessment Report of the Intergovernmental Panel on Climate Change, the influence of human activities has warmed the climate in an unprecedented way over the past 2000 years, resulting in widespread and rapid changes in the earth’s atmosphere, oceans, and cryosphere. Under the pressure of such a severe climate change situation, many governments have begun to formulate relevant policies and take action to try their best to reduce greenhouse gas emissions. The United States, Sweden, Canada, and other countries have reduced carbon emissions by imposing carbon taxes, formulating energy efficiency standards for electrical appliances, and establishing markets for carbon dioxide [5,6]. China has implemented carbon emission reduction measures such as a carbon emissions trading system, energy price reform, vehicle fuel economy standards, and clean energy technology deployment, and is currently a world leader in solar and wind energy manufacturing and use technologies. In addition, the formulation and implementation of new environmental agreements with basic technology diffusion rules, innovation of environmental technologies, and exchanges and cooperation between regions are also important measures to reduce the greenhouse effect [4].

With the acceleration of urbanization and industrialization, the social economy and residents’ living standards have been greatly improved. Residents are the main consumers of industrial and agricultural products and services, and energy consumption and carbon dioxide emissions from residents have a great contribution to the total energy consumption and carbon dioxide emissions of each country. According to statistics, 72% of the world’s total carbon dioxide emissions come from residential consumption. More than 80% of energy use and carbon emissions in the United States come from residential consumption and related economic activities. Britain’s residential carbon emissions account for 74% of total emissions. In China, carbon dioxide emissions from residential consumption account for 40% to 50% of the total emissions, and it has become the second largest energy consumer after industry [7]. Residential carbon emissions are divided into two categories: direct carbon emissions and indirect carbon emissions. The carbon dioxide emissions generated by the direct consumption of fossil energy sources by residential buildings, electricity, cooking, and transportation in the process of residents’ living are called direct residential carbon emissions. The carbon emissions generated by indirect energy consumption in the production, transportation, use, and operation of food, daily necessities, and services related to residents’ lives are called residents’ indirect carbon emissions [8,9]. It can be seen that it is of great significance for slowing down global climate change and accelerating the achievement of carbon peaking and carbon neutrality by clarifying the current status and future trends of carbon dioxide emissions in residents’ activities and taking corresponding carbon emission reduction control measures according to local conditions [10,11].

There have been many research results on residential carbon emissions. The research focus involved residential clothing, food, housing, transportation, and services, mainly using mathematical models and computer software to quantitatively measure household consumption carbon emissions [12], household energy demand [13], transportation carbon emissions [14], and residential building carbon emissions [15,16]; scholars combined these with relevant theories to discuss the driving factors of residential carbon emissions [17] and residential low-carbon consumption willingness [18]. These studies scientifically revealed the contribution rate and impact of personal behavior, energy use, and government macro policies on residential carbon emissions. With the rapid development of computer software, information visualization technology has been widely used in scientific research, such as knowledge theory, paradigm structure, and trend evolution [19]. However, there is currently a lack of overall review and integration of research hotspots, research trends, and development contexts in the field of residential carbon emissions.

Based on the above background, we tried to sort out and integrate the related literature on residential carbon emissions included in the Web of Science and CNKI through the CiteSpace 5.8.R1 software (Professor Chaomei Chen from Institute of Computer and Information Science, Drexel University, Philadelphia, PA, USA) and aimed to combine it with a visual knowledge map to analyze the research status and research hotspots of residential carbon emissions from a macro perspective and provide a reference for follow-up research and other scholars to understand the development context and research trends of residential carbon emissions. The remainder of this article is organized as follows: the second part of this paper introduces the data sources and research methods used in this study; the third part introduces the current status characteristics of residential carbon emissions research, including annual changes in the number of publications, author collaboration networks, and citations of papers; the fourth part analyzes the hotspots of residential carbon emissions research and how they evolve. Finally, in the fifth part, we summarize the main research findings and future directions. Figure 1 is the research framework of this paper.

## 2. Data Sources and Analytical Methods

### 2.1. Data Sources

This paper used the Web of Science core collection and the China National Knowledge Infrastructure database as the primary data sources. The selected period was from 1 January 2000 to 31 December 2021. In the Web of Science database, we used “resident CO_2_ emission” and “resident carbon emission” as the subject words to guide the literature search. Then, we selected the literature type as “article” and the database as “Web of Science core collection”, excluded publications related to toxicology, chemistry, medicine, and other unrelated fields, checked the remaining publications’ titles one by one, and deleted irrelevant publications. Finally, 480 related studies were screened. In the China National Knowledge Infrastructure database, we used “residential carbon emission” as the subject word to search for relevant literature, and in the literature source category we selected “Engineering Index Source Journals”, “Chinese Social Science Citation Index”, “Core Journals”, and “China Science Citation Database”, and 548 Chinese studies were screened. Finally, a total of 1028 publications were used as the primary data for our research. All this literature focused on residential carbon emissions and on one or more regions to conduct relevant research.

### 2.2. Analytical Methods

CiteSpace software is a Java-based document information analysis software developed by the research team of Professor Chen from Drexel University in the United States and the Dalian University of Technology in China. It can delve deep into and visualize the internal connections between research objects, revealing the frontier hotspots and evolution trends in the research field. The software has been widely used in ecology, economics, medicine, and other fields [20,21,22]. We used the literature data on residential carbon emissions as the research data. We used the tools of co-occurrence analysis, outburst detection, cluster analysis, and other tools of CiteSpace 5.8.R1 software to draw the visual knowledge networks about authors, publishing institutions, subject terms, and references.

Moreover, we visualized the publication trend of residential carbon emission research, primary research authors and research teams, and literature citations. We identified frontier research and hot topics in different development stages. Through these analyses, we described the development trend of residential carbon emissions research. We predicted future research aspects that may need to be focused on.

## 3. Basic Situation Analysis

### 3.1. Trends in the Number of Publications

We analyzed 1035 publications published between 2000 and 2021. During this research period, the number of publications related to residential carbon emissions showed an overall increasing trend year by year, roughly divided into three development stages, and the results are shown in Figure 2.

The period from 2000 to 2008 was the initial stage of residential carbon emission research. The number of publications in this stage only accounted for 2.22% of the total, and there were less than 10 publications in any year. During this period, scholars from the United States and the United Kingdom first researched the impact of transportation carbon emissions on human health, the relationship between energy consumption and carbon emissions, and the measurement of industrial activities and solid waste carbon emissions [23,24,25,26,27]. These studies laid a solid foundation for subsequent research. 

The period from 2009 to 2014 was the rising stage of residential carbon emission research. The number of publications increased exponentially, 12.70 times that of the previous period, accounting for 28.21% of the total. During this period, the research topics and methods were gradually enriched with the in-depth cooperation of multiple departments and fields. Scholars worldwide discussed the relationship between residents’ behavior [28,29], residential buildings [30,31], household energy consumption [32], urban form and commuting patterns [33,34], residents’ leisure tourism [35,36], and other aspects, with carbon dioxide emissions. There were also studies evaluating the effectiveness of existing environmental climate policies [37,38]. 

Since 2015, residential carbon emission research has reached a prosperous stage, and the number of publications has increased significantly, with more than 100 papers published in the past four years. Especially after the Paris Climate Change Conference in 2015, research such as how to deal with climate change and how to reduce residential carbon emissions has attracted the attention of more national governments and scholars [11,12,13,14,15,16]. Regarding literature source countries, China, the United States, and the United Kingdom rank in the top three in the number of publications on the Web of Science.

### 3.2. Cooperation Networks

We used the author co-occurrence analysis tool of the CiteSpace software to conduct a visual knowledge graph analysis of the primary authors and their cooperation in residential carbon emissions. It can be seen from Figure 3 that the results of the author’s cooperation network that the research teams studying residential carbon emissions are relatively scattered, and the degree of connection between the teams is low. According to the statistics on the number of publications, the top 10 scholars published 162 papers, accounting for 15.76% of the total papers. The research institutions of the publications were mainly from China, such as Lanzhou University, China University of Mining and Technology, Henan University, Xi’an University of Architecture and Technology, Wuhan University, and Fudan University, and the number of publications in the first four research institution was more than 10. Professor Peng’s team from Fudan University and professor Qu’s team from Lanzhou University were the first to research residential carbon emissions in China. They have successively built a model on the impact of family patterns, population, and household consumption on carbon emissions in China, quantitatively measured residential living energy and consumer goods carbon emissions [39,40,41,42], and analyzed fixed carbon emissions from residential buildings [43]. The teams of Chen Hong, Long Ruyin, and Wei Jia from the China University of Mining and Technology were mainly engaged in the research on carbon emission measurements of rail transit [44], analysis of residents’ carbon capacity and influence factors [45], and residents’ response and evaluation of carbon emission reduction policies [46]. In addition, scholars from research institutions such as the University of Iceland, University College Dublin in Ireland, and Hamad Ben Khalifa University have also conducted in-depth research on residential carbon emissions. Among them, the research team composed of Heinonen Jukka, Otten Juudit, Arnadottir Arora, and other scholars from the University of Iceland has studied the relationship between residents’ behavior and greenhouse gas emissions and energy consumption [47], the impact of low-energy residential buildings on climate change [48], the effectiveness of carbon emission reduction strategies [49], the estimation of urban residents’ consumption carbon footprint [50], the correlation analysis between travel patterns and carbon emissions [51], and has published many papers in excellent journals.

### 3.3. Frequently Cited Literature and Source Analysis

The most frequently cited publications are foundational and vital to research. We searched the Web of Science core collection for related literature on residential carbon emissions and sorted out the top 10 most cited literature from 2000 to 2021, as shown in Table 1. The most cited literature is “A high-resolution domestic building occupancy model for energy demand simulations” published in the journal *Energy and Buildings* by Richardson, Infield, and other scholars of Loughborough University, U.K., and the number of citations is 310. Based on data from a survey detailing residents’ behavior throughout the day, the research used Markov chain technology to simulate residential activity and energy use in U.K. households [28]. The second most cited article was “The health risks and benefits of cycling in urban environments compared with car use: health impact assessment study” published in the *British Medical Journal* in 2011. The research modeled the relationship between Barcelona residents’ road traffic travel patterns, road traffic accidents, and carbon dioxide emissions to assess the impact of the region’s bike-sharing program on residents’ health [52]. The paper “Cities and greenhouse gas emissions: moving forward” written by Daniel, the World Bank’s chief urban expert in Washington, D.C., also has many citations. This research analyzed the characteristics of per capita greenhouse gas emissions, GDP, solid waste emissions, and other characteristics of the member countries of the Intergovernmental Panel on Climate Change, and pointed out a strong correlation between regional greenhouse gas emissions and solid waste generation [53]. The results of the literature sources are shown in Figure 4. On the Web of Science website, the journals of the top five number of publications are the *Journal of Cleaner Production*, *Sustainability*, *Energy Policy*, *Science of The Total Environment*, and *Environmental Science and Pollution Research*, and the total number of papers published in these five journals accounted for 34.17% of the total number of papers published in Web of Science.

## 4. Basic Situation Analysis

The keyword is a high-level summary and condensation of the research content and theme of the entire paper. Clustering analysis of keywords with high frequency and analysis of the trend of keywords can accurately grasp the hotspots and development trends in the research field. We used the co-occurrence, emergence, and clustering analysis tools of the CiteSpace software and set the analysis node as the keyword. The time interval was one year to draw the clustering and emergence map of the keywords of residential carbon emission publications.

### 4.1. Research Hotspots

Through the co-occurrence frequency statistics of the keywords of residential carbon emission publications, we found that the keywords with high co-occurrence frequency were carbon emission, impact, energy consumption, climate change, behavior, residential consumption, urbanization, Stirpat model, carbon reduction emissions, and low-carbon economy (Figure 5). We used the keyword clustering analysis tool based on the LLR algorithm in the CiteSpace software to perform clustering operations on keywords to further explore the hotspots of residential carbon emissions research (Table 2). The results showed that the research on residential carbon emissions mainly focused on climate change and low-carbon urban development research, quantitative measurement of direct and indirect carbon emissions such as residents’ consumption, transportation, and tourism, surveys on residents’ low-carbon willingness, and analysis of factors affecting carbon emissions.

#### 4.1.1. Climate Change and Low-Carbon Urban Development

Clusters #0, #2, #4, and #11 were climate change, urban form, air pollution, and low-carbon economy, respectively, and these clusters mainly described the research background of residential carbon emissions. In 1988, the World Meteorological Organization and the United Nations Environment Programme established the Intergovernmental Panel on Climate Change to regularly assess the potential impacts and future risks of global climate change and provide scientific advice for governments to formulate climate change-related policies. In 1992, the United Nations Conference on Environment and Development promulgated the Framework Convention on Climate Change, calling on developed countries to take corresponding measures to limit anthropogenic greenhouse gas emissions. In December 2015, many countries signed the “Paris Agreement” at the United Nations Climate Change Conference. They pledged to take action towards the goal of “striving to limit the rise in global average temperature to less than 1.5 °C compared with the industrialization period” [4]. Since then, climate change research has become a hot topic of concern to government organizations and scholars worldwide. Relevant studies showed that climate change will not only hinder regional social and economic development [54,55], but also destroy the stability of natural ecosystems to a certain extent, resulting in a reduction in biodiversity [56], atmospheric environmental pollution [57], change of land use activities [58], reduction of food production [59], and other phenomena, and even pose a serious threat to human health [60,61].

Since the 21st century, social economy and urban construction have achieved rapid development. Cities are the main gathering places for human activities. In cities, energy supply, transportation, waste management, and infrastructure construction are closely related to greenhouse gas emissions. With the continuous growth of the urban population, vicious environmental problems such as air pollution and lack of resources will become more serious. Developing a low-carbon economy and low-carbon technology is the key to improving urban environmental problems [62]. In 2003, the UK proposed to promote social development and improve energy efficiency by establishing a low-carbon economic model with low energy consumption, low pollution, and low emissions, and promulgated the world’s first climate change law, the Climate Change Act, which provided adequate legal protection for the development of a low-carbon economy [63]. Since then, experts and scholars around the world have actively explored the connotation and effective paths of low-carbon economic development and low-carbon city construction. In the Eleventh Five-Year Plan, the Chinese government pointed out that improving resource utilization efficiency should be one of the main goals of social and economic development and should actively promote the construction of a resource-saving and environmentally friendly society, and urban low-carbon transformation can be effectively promoted by stimulating urban green innovation and optimizing urban resource allocation [64]. The California state legislature has formulated the state’s response to the global warming bill, which proposed measures to increase the efficiency of renewable energy power systems and waste recycling rates and requires each metropolitan area to take sustainable community strategy, regional housing needs assessment, traffic planning, and greenhouse gas emissions reduction targets to reduce greenhouse gas emissions [65]. By discussing the relationship between low-carbon goals and urban spatial planning, British scholars pointed out that new technologies should be used to reduce building energy consumption and promote the development of a low-carbon society in the UK [66].

#### 4.1.2. Quantitative Calculation of Residential Carbon Emissions

Clusters #1, #6, #8, #9, and #10 are sustainability, carbon emissions, Beijing, residential consumption, and household carbon emissions, respectively, and these clusters mainly focus on the quantitative measurement of residential carbon emissions. The carbon dioxide emissions generated by the direct consumption of fossil energy by residential buildings, electricity, cooking, and transportation in the process of residents’ living are called direct residential carbon emissions. The carbon emissions generated by the indirect energy consumption in the production, transportation, use, and operation of food, daily necessities, and services related to residents’ lives are called residents’ indirect carbon emissions [8,9,67]. Many scholars have conducted in-depth research on this problem from various angles using the input–output model, exponential decomposition model, and other mathematical models.

(1)Residential Living Carbon Emissions.

Electricity, cooking, heating, and other behaviors closely related to the daily life of residents, as well as the disposal of household waste and the construction of residential buildings, will emit greenhouse gases into the atmosphere and are all carbon emissions from residences [68]. According to relevant data, the proportion of carbon emissions from the construction industry in many countries in the world has reached more than 40% of the total carbon emissions from energy consumption, and the construction industry has gradually become the largest energy consumption industry. It is an inevitable trend for the development of the construction industry to explore low-carbon technologies and reduce residential energy consumption [69,70,71,72]. Scholars from Greece, Serbia, Kazakhstan, and other countries have combined the data of residential buildings such as energy consumption, material use, building environment, air quality, and operational security to construct sustainable performance evaluation models for residential buildings through scenario simulation and mathematical models, and pointed out that the construction industry needs to improve the waste management system to promote the ecological sustainability of residential buildings [15,73,74]. With the improvement of the level of science and technology, more and more new technologies and clean energy have penetrated family life. Taking measures such as building energy-saving technologies and using renewable energy-powered supply systems can effectively alleviate the carbon emissions of family living [75]. In addition, the level of regional social economy and residents’ income are the main reasons for the differences between urban and rural residential buildings’ carbon emissions [16,76].

(2)Transportation and Tourism Carbon Emissions.

With the development of the social economy and the improvement of residents’ living standards, the demand for automobiles has gradually increased. The carbon emissions generated by transportation have also increased yearly. In 2009, carbon emissions from transportation accounted for more than 20% of the total global carbon emissions, which was the third source of carbon emissions after industry and construction. The construction of low-carbon green transportation has become an important strategy for mitigating climate change worldwide. In the 13th Five-Year Plan, the Chinese government emphasized the need to reduce the transportation sector’s carbon emission intensity and energy consumption by optimizing the urban transportation structure and guiding residents to change their travel modes. New Zealand scholars pointed out that shared transportation can effectively reduce energy consumption and bring good economic dividends to the local society and suggested using emerging technologies such as the Internet to promote the planning, construction, and development of public transportation [9]. While tourism has contributed significantly to society and the economy, it has also increased greenhouse gas emissions to a certain extent. According to some estimates, the carbon emissions from tourism account for 8% of the total global greenhouse gas emissions and will continue to grow at a rate of 2.5% every year, which will undoubtedly put great pressure on the environment. Transport is a major source of carbon emissions from tourism. In addition, energy intensity, tourism consumption level, tourism scale, and population size are the factors affecting the difference in tourism carbon emissions, among which tourism consumption level and energy intensity contribute the most [77].

(3)Indirect Carbon Emissions of Residents.

At present, household consumption demands and residents’ consumption patterns have undergone tremendous changes. Since the 1990s, household energy consumption has exceeded industrial energy consumption. The indirect energy consumption and carbon dioxide emissions generated by household consumption in China were increasing yearly, and residents’ consumption behavior shifted from survival to self-improvement and leisure [78]. Based on models such as the input–output method, life cycle assessment method, and consumer lifestyle method, scholars calculated and compared the different characteristics of indirect carbon emissions from household consumption under different research scales such as national and provincial levels [79,80].

It can be seen from the above review that the goal orientation and research content of the quantitative research on residential carbon emissions are very rich, and the research scales include county, city, provincial, and national levels. The research method has gradually shifted from early qualitative research only for policy discussion to quantitative research using mathematical model evaluation such as the coefficient and input–output methods. These studies accurately reflect the current situation and trend of regional carbon emissions. They can provide an objective and scientific reference value for the region and the world to formulate relevant policies to deal with climate change and low-carbon emission reduction.

#### 4.1.3. Analysis of Influencing Factors of Residential Carbon Emissions

Clusters #5, #7, and #12 are influencing factors, Stirpat model, and scenario building, respectively, and these clusters mainly discuss the influencing factors of residents’ carbon emissions. Exploring the impact mechanism of residential carbon emissions is critical in effectively mitigating greenhouse gas emissions and addressing climate change. Scholars usually use structural decomposition, the Stirpat model, exponential decomposition, and other model methods for quantitative analysis. The research found that factors such as urbanization development level, residents’ income, population structure, energy structure, consumption structure, and technological development have a specific impact on residential carbon emissions. Household income was positively correlated with consumer energy consumption and carbon emissions [81], the urban–rural income gap was negatively correlated with indirect carbon emissions [82], and the contribution of electricity carbon emissions in the housing energy system accounted for 64–73% [83].

#### 4.1.4. Analysis of Residents’ Low Carbon Willingness

Cluster #3 is the theory of planned behavior, which focuses on the research on residents’ low-carbon consumption willingness. Human beings are the main terminal body of consumption. An in-depth understanding of human consumption behavior, environmental protection attitude, energy demand, and policy awareness has essential reference value for the development of low-carbon products and services and the implementation of energy-saving and emission-reduction strategies. The theory of planned behavior is one of the widely used theories in studying human behavior [84]. In recent years, the research on residents’ low-carbon behavior and intention based on this theory has gradually increased, mainly around the family income and energy prices, family population, lifestyle, society, culture, and personal factors such as environmental protection consciousness. They found that residents’ energy-saving attitudes, family income, and level of education will positively influence energy saving [18,85,86,87]. Among them, residents’ attitude toward energy saving has a more significant impact on low-carbon consumption behavior [88]. In contrast, factors such as non-monetary incentive policies and measures have a minor impact on residents’ intention to save energy [89].

### 4.2. Research Trends

We used CiteSpace software to perform sudden detection of keywords; the results are shown in Figure 6, and the thematic evolution and technical methods of residential carbon emissions in each period are summarized.

#### 4.2.1. Evolution of Research Topics

Through the emergent detection tool of CiteSpace software, we performed an emergent analysis of the keywords of the publications, and we summarized the evolution process of the research topic in each research period.

(1)From 2006 to 2013, the emergent words included climate change, ecological footprint, structural decomposition analysis, and input–output analysis, among which climate change appeared first and lasted the longest. Since the 20th century, developed regions such as the United States and the European Union were the first to take climate change research as a critical concern in the environmental field, and formulated measures such as renewable energy development to reduce greenhouse gas emissions, and relevant government departments at all levels responded positively. Climate governance was gradually linked to social and resident activities such as urban infrastructure construction, commercial activities, community development capabilities, and household energy consumption [90,91,92], and quantitative analysis of carbon footprints has been gradually applied to the research of energy and electricity consumption, and at the residential, commercial, and other aspect levels [29,93]. China’s research on residential carbon emissions started later than other countries. During this period, scholars mainly focused on quantitative measurement and prediction of carbon emissions from energy consumption, analysis of influencing factors of household energy consumption, and the impact degree of household patterns on carbon emissions. Methods such as questionnaire surveys, comprehensive life cycle method, structural decomposition method, and input–output method were applied in the research [94,95,96,97,98,99].(2)From 2014 to 2018, the emergent words included architecture, transportation, urban residents, the Stirpat model, urbanization, and indirect carbon emissions, among which the emergent time of transportation was the longest. The successful convening of the 21st United Nations Climate Conference during this period raised the attention of the governments of member states to the research on climate change and carbon dioxide emissions to a new height. Scholars’ research perspectives and methods were more abundant, and the research content mainly focused on the aspects of “living” and “travel” of residents. With the rapid development of the economy, greenhouse gas emissions related to transportation have increased rapidly. Scholars from the United States, Finland, New Zealand, and other countries have analyzed and compared the different characteristics of carbon emissions caused by different transportation means and advocated that low-carbon transportation should be used as an essential measure to implement carbon emission reduction policies [8,77,100,101]. The close cooperation between regions has also made tourism gradually become the first choice for residents to relax, increasing the local economic revenue and bringing many environmental and climate problems. Scholars from Germany, Spain, Finland, Sweden, and other countries have successively studied the carbon emissions of long and short trips from the perspectives of living location, travel destination, and travel mode and discussed the correlation between population, social economy, and travel carbon emissions [102,103]. The *2017 World Energy Outlook* stated that global residential energy demand was expected to grow by 32% due to population growth, which will undoubtedly exacerbate the environmental crisis [88]. New low-energy buildings are one of the feasible ways to reduce greenhouse gas emissions. After the European Union, the Netherlands, and other countries implemented a series of legislative actions to promote the construction of low-energy buildings, some scholars used qualitative and quantitative analysis methods to conduct relevant research on the carbon emissions of building energy consumption [104]. During this period, especially after the 19th National Congress of the Communist Party of China, China’s urbanization construction achieved remarkable results. Scholars began to pay attention to the impact of urbanization development, population structure, and urban and rural consumption structure on carbon emissions, quantitative measurements, and driving factors of direct and indirect household carbon emissions [105,106,107].(3)From 2019 to 2021, the emergent words included planned behavior and carbon emission reduction theory. During this period, the research on residents’ low-carbon consumption behavior intention and its influencing factors had become a popular research topic. Scholars from Poland, Pakistan, China, and other countries have provided new ideas for mobilizing the public to participate in low-carbon consumption behavior by quantitatively analyzing the driving mechanism of residents’ willingness to consume low-carbon from the perspectives of personal cognition, product perception, household income, and incentive policies [86,87,89].

In general, since the 21st century, the research scale of residential consumption carbon emissions has been broad, involving the macro-scale of national, provincial, and municipal levels and the micro-scale of community. The research mainly focuses on quantitatively analyzing direct and indirect carbon emissions of residential buildings, transportation, and tourism. The proposal of emerging policies has further raised the attention of research on the influencing factors of urbanization development and social economy on residential carbon emissions and low-carbon willingness to a new height.

#### 4.2.2. Data Sources and Technical Methods

Statistical and spatial data are used in the study of residential carbon emissions. The statistical data include population size, age composition, household income, consumption structure, living environment, economic level, urbanization rate, industrial structure, technological level, energy structure, and other data, mainly from statistical yearbooks of governments at all levels, official statistics, and field questionnaires. These traditional statistics are slower to update and less accurate. Data acquisition methods have become more diverse with the vigorous development of technologies such as computers, artificial intelligence, remote sensing, and geographic information systems. Remote sensing images obtained by high-precision satellites can extract long time series of land use cover, vegetation coverage, night light index, and other data. Communication big data such as mobile phone signaling can extract residents’ consumption and other behavioral information with spatiotemporal markers. The combination of these data and traditional statistical data has dramatically improved the accuracy and spatiotemporal dynamics of the research on residential carbon emissions.

The diversified application of research data has promoted the evolution of the methods and technical means of residential carbon emission research towards modernization and diversification. Existing residential carbon emissions research mainly use mathematical models, with less application of computer technology and artificial intelligence technology. Based on all kinds of statistical data collected by questionnaire survey and online queries, combined with mathematical models such as the input–output model, consumer lifestyle method, factor decomposition method, Stirpat model, structural equation model, residential carbon emissions measurement, and influencing factor analysis can be carried out. Using ENVI, ArcGIS spatial econometric analysis software technology can realize the related research data of spatial information extraction, dynamic visual display, and spatial difference analysis. In addition, some scholars use the system dynamics model for dynamic simulation of residential carbon emissions to predict the future trend and the related factors on the impact of carbon emissions. The application of these intelligent technical means can significantly improve the spatial dynamics of residential carbon emission research, but it has not yet become a current research hotspot. Future methods such as spatial analysis and artificial intelligence can be further applied to research residential carbon emissions.

## 5. Conclusions and Prospect

### 5.1. Conclusions

In this study, based on the Web of Science and CNKI literature databases, we used CiteSpace 5.8.R1 visualization knowledge graph software to review and comprehensively integrate the research status, research hotspots, and research trends of residential carbon emissions from a macro perspective. The development context and evolution trend of this research field in the past two decades were also explored, which had specific reference value for the future research on residential carbon emissions. The main conclusions are as follows:(1)Through a comprehensive analysis of the number of documents, authors, and document sources, we found that during the study period, the number of published papers on residential carbon emissions research generally increased year by year. Preliminary research by scholars in the United States and the United Kingdom laid the foundation for the development of the field of residential carbon emissions. The co-occurrence network of the authors of the article showed that the current research teams related to residents’ carbon emissions were relatively scattered, and the degree of connection between the teams was low.(2)We found that research hotspots on residential carbon emissions have been constantly evolving, roughly experiencing three research stages of initial, rising, and prosperous in the past two decades. Research hotspots mainly focus on policy analysis on climate change and energy demand, quantitative measurements, and impact mechanism analysis of residents’ direct and indirect carbon emissions, and research on residents’ willingness to consume low carbon levels.(3)The development of computer technology has enriched the research perspectives and technical means of residential carbon emissions. The research content has gradually turned to quantitative analysis of residents’ direct and indirect carbon emissions, such as residential buildings, transportation, and tourism, relying on big data and mathematical models. The proposal of emerging policies has further raised the attention to the impact of urbanization development and social economy on residential carbon emissions to a new height.

### 5.2. Prospects

An in-depth discussion of residential carbon emission sources, driving mechanisms, and regional differences are of great significance for formulating environmental management and control strategies and establishing low-carbon consumption models. This paper used bibliometric analysis software to sort out relevant literature data and analyzed the general situation and hot trends of residential carbon emissions research from a macro perspective. In general, the research system on household consumption, transportation, leisure tourism, and direct and indirect carbon emissions is relatively mature. The selection of indicators and methods has good representativeness and rationality. The research results can accelerate the pace of building a low-carbon society to a certain extent and provide a good reference for environmental protection, meteorological, urban planning, and other departments. Future research on residential carbon emissions needs to focus on the following aspects:(1)The proposal of carbon peaking and carbon neutrality targets puts the research on residential carbon emissions in a critical period of prosperity and development. The communication and cooperation between research teams should be strengthened, the knowledge and theories of geography, sociology, ecology, artificial intelligence, and other disciplines should be integrated, and new research perspectives should be explored from the actual situation. Most of the current research takes developed countries as case areas, but the conflicts between social and economic development, environmental protection, and residents’ lives in developing countries are becoming more serious. Therefore, the research on residential carbon emissions in developing countries should be strengthened to alleviate the pressure of global carbon emission reduction.(2)At present, most of the research in this field uses mathematical models to analyze relevant statistical data. However, remote sensing technology, geographic information system technology, artificial intelligence technology, and other modern computer methods are less used in the field of residential carbon emission. These intelligent technologies can improve the dynamics and accuracy of residential carbon emission research. Future research can be based on environmental and spatial big data such as remote sensing ecological index, night light data, etc., flexibly using spatial analysis technologies such as remote sensing and geographic information system, and strengthening the comparison of regional differences in residential carbon emissions, spatiotemporal dynamic evolution, trend prediction, and other related studies.(3)Relevant studies showed that urbanization development, regional economic development, unbalanced development of technological innovation levels, residents’ intentions, and household consumption patterns were the influencing factors that lead to significant differentiation of residential carbon emissions. Therefore, we suggest that national government departments and international organizations should actively explore new low-carbon economic development policies, increase support for science and technology innovation in economically backward regions, strengthen environmental technology exchange and cooperation between regions, further improve support policies and regulatory systems for developing countries, and guide household consumption behavior by formulating incentive mechanisms for green consumption and popularizing scientific knowledge about energy conservation and emission reduction.

## Figures and Tables

**Figure 1 ijerph-20-01706-f001:**
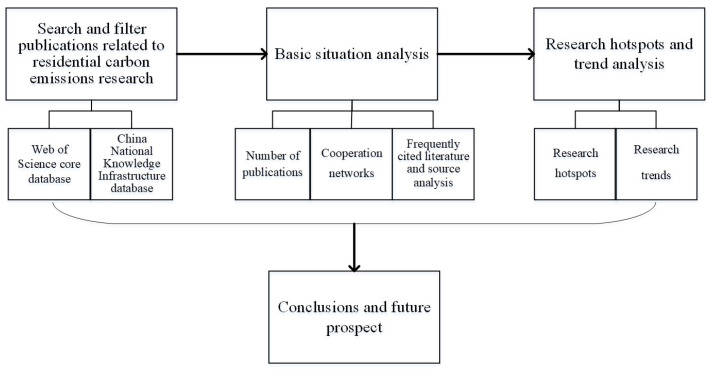
The research framework of this paper.

**Figure 2 ijerph-20-01706-f002:**
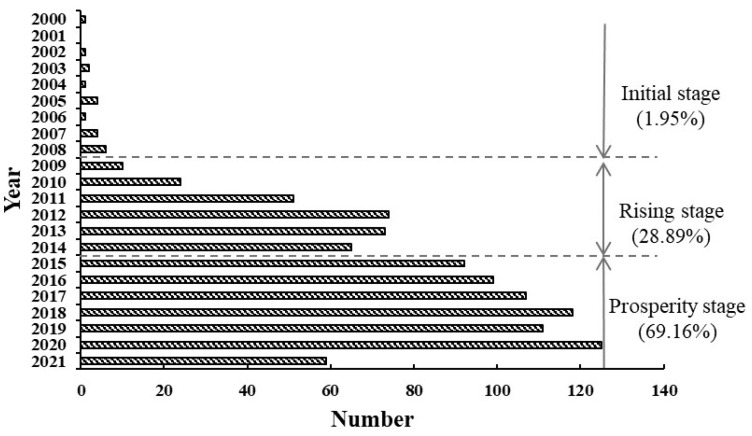
Annual changes in the number of residential carbon emissions publications from 2000 to 2021.

**Figure 3 ijerph-20-01706-f003:**
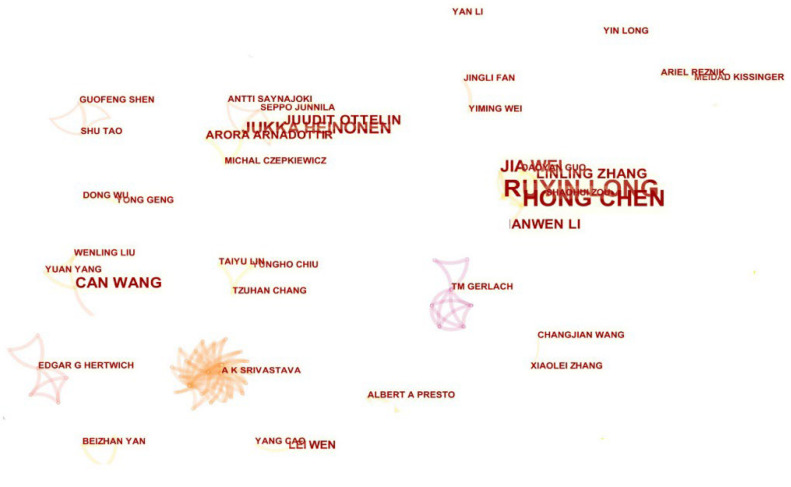
Map of a collaborative network of authors in the field of resident carbon emission.

**Figure 4 ijerph-20-01706-f004:**
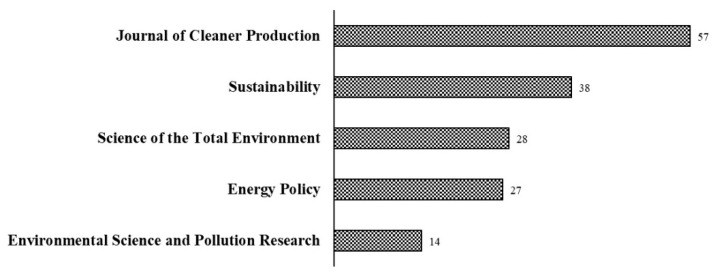
Top five journals by the number of publications on Web of Science.

**Figure 5 ijerph-20-01706-f005:**
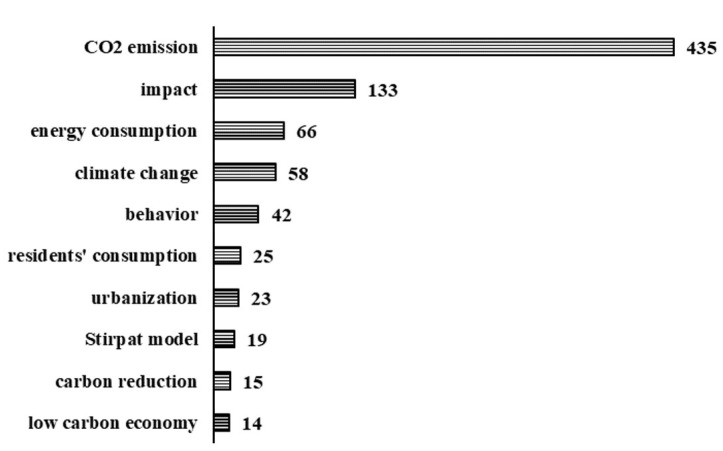
The frequency of keywords in publications.

**Figure 6 ijerph-20-01706-f006:**
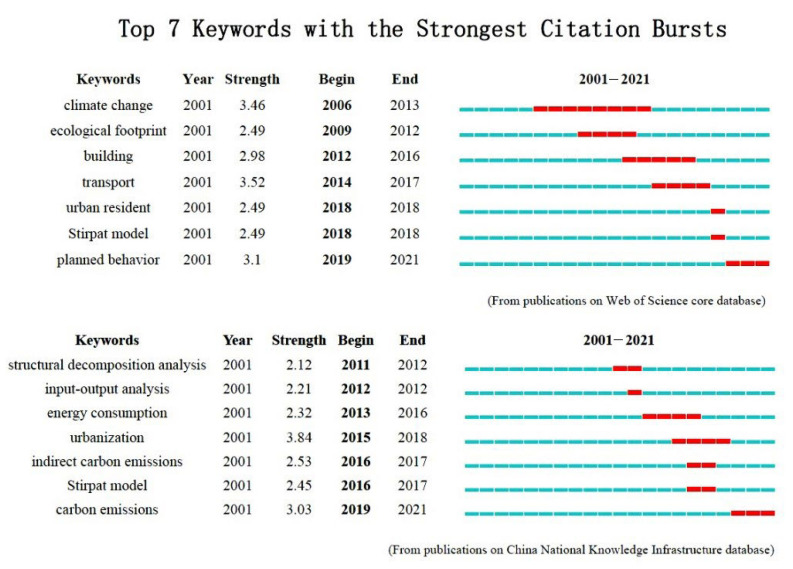
Clustering map of keywords of Web of Science publications.

**Table 1 ijerph-20-01706-t001:** Top 10 papers of cited number from 2000 to 2021 in Web of Science.

Paper Title	Authors	Year	Journal	Number of Citations
A high-resolution domestic building occupancy model for energy demand simulations	Richardson Ian, Thomson Murray, Infield David	2008	*Energy and Buildings*	310
The health risks and benefits of cycling in urban environments compared with car use: health impact assessment study	Rojas-Rueda David, de Nazelle Audrey, Tainio Marko, et al.	2011	*British Medical Journal*	243
Cities and greenhouse gas emissions: moving forward	Hoornweg Daniel, Sugar Lorraine, Gomez Claudia, et al.	2011	*Environment and Urbanization*	239
The impact of lifestyle on energy use and CO_2_ emission: An empirical analysis of China’s residents	Wei Yiming, Liu Lancui, Fan Ying, et al.	2007	*Energy Policy*	201
The work of policy: actor networks, governmentality, and local action on climate change in Portland, Oregon	Rutland Ted, Aylett Alex	2008	*Environment and Planning D-society & Space*	158
In-field greenhouse gas emissions from cookstoves in rural Mexican households	Johnson Michael, Edwards, Rufus, Frenk, Claudio Alatorre, et al.	2008	*Atmospheric Environment*	132
How much transport can the climate stand? Sweden on a sustainable path in 2050	Akerman Jonas, Hojer Mattias	2006	*Energy Policy*	108
Contributing to local policy making on GHG emission reduction through inventorying and attribution: A case study of Shenyang, China	Xi Fengming, Geng Yong, Chen Xudong, et al.	2011	*Energy Policy*	94
A low-carbon scenario creation method for a local-scale economy and its application in Kyoto city	Gomi Kei, Shimada Kouji, Matsuoka Yuzuru, et al.	2010	*Energy Policy*	89
CO_2_ and its correlation with CO at a rural site near Beijing: implications for combustion efficiency in China	Wang Y, J W Munger, Xu S, et al.	2010	*Atmospheric Chemistry and Physics*	89

**Table 2 ijerph-20-01706-t002:** Cluster analysis of keywords.

Clustering Number	Clustering Name	Subcluster Name	Source of Keywords
#0	climate change	CO_2_ emission, energy consumption, China, etc.	publications in Web of Science database
#1	sustainability	emission, carbon footprint, resident, etc.	
#2	urban form	greenhouse gas emission, city, transport, etc.	
#3	theory of planned behavior	model, performance, building, etc.	
#4	air pollution	climate change, policy, energy efficiency, etc.	
#5	scenario building	impact, consumption, urban, etc.	
#6	carbon emission	Urbanization, energy consumption, energy conservation and emission reduction, etc.	publications in China National Knowledge Infrastructure database
#7	influence factor	Imdi model, low carbon development, built environment, etc.	
#8	Beijing city	low-carbon city, transportation carbon emissions, residential families, etc.	
#9	residential consumption	urbanization, energy consumption, input–output analysis, etc.	
#10	household carbon emissions	carbon emission reduction, population structure, living consumption, etc.	
#11	low carbon economy	carbon emission intensity, low carbon consumption, rural residents, etc.	
#12	Stirpat model	demographic factors, consumption patterns, spatiotemporal differences, etc.	

## Data Availability

Not applicable.

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
