# Peer review of "Research Progress and Hotspot Analysis of Residential Carbon Emissions Based on CiteSpace Software"

_ijerph, 2023, doi:10.3390/ijerph20031706_

Round 1

Reviewer 1 Report

Research progress and hotspot analysis of residential carbon emissions based on CiteSpace software

Summary

The topic of the paper it is very interesting.  The author propose that the future research direction of residential carbon emissions should focus more on spatial analysis and trend prediction based on intelligent methods under a low-carbon background.

Minor Revisions:

In hierarchical terms, a native speaker should review the English language.

Formatting should be revised with respect to Journal standards

Detailed comments

1.      In the introduction, I suggest making changes respecting the following points:

a.       Discuss more generally the issues concerning greenhouse gas emissions.

b.      Discuss how international climate agreements should facilitate local actions in climate protection

c.       Quote how international cooperation can foster local cooperation in reducing greenhouse gas emissions. (i.e. Cooperation, diffusion of technology and environmental protection: a new index. Written by Barra, Bimonte, Senatore. Quality & Quantity (2019) 53:1913–1940)

2.      In the conclusions the authors must write more clearly and in more detail the results obtained as useful tools for future research developments on this topic, especially in points 2 and 3

Author Response

Response to Reviewer 1 Comments

Point 1: In the introduction, I suggest making changes respecting the following points: a. Discuss more generally the issues concerning greenhouse gas emissions. b. Discuss how international climate agreements should facilitate local actions in climate protection. c. Quote how international cooperation can foster local cooperation in reducing greenhouse gas emissions. (i.e. Cooperation, diffusion of technology and environmental protection: a new index. Written by Barra, Bimonte, Senatore. Quality & Quantity (2019) 53:1913–1940)

Response 1: Thank you very much for your suggestions. We have addressed your comments and revised the introduction. In the first paragraph of the introduction, we added a description of the greenhouse effect and greenhouse gases. In the second paragraph of the introduction, we reviewed the efforts of climate agreements promulgated by international organizations in reducing CO2 emissions and the responses of countries around the world. The reference you suggested was quoted, and the significance of international environmental technological innovation and cooperation to reduce CO2 emission was added. See the “1. Introduction” of the paper for details.

Point 2: In the conclusions the authors must write more clearly and in more detail the results obtained as useful tools for future research developments on this topic, especially in points 2 and 3.

Response 2: Thank you very much for your suggestions. We redivided the conclusions into two sections of conclusions and prospect. In section 5.1, we have made the main conclusions clearer by writing them in points. In section 5.2, we added the description of the research trend of future residents' carbon emissions, and put forward suggestions on the national response to low-carbon development. See the “5. Conclusions and Prospect” of the paper for details.

Minor revisions: In hierarchical terms, a native speaker should review the English language. Formatting should be revised with respect to Journal standards.

Response: Thank you very much for your suggestions. We have checked and revised the grammar and format of the paper based on the journal standards. See the paper for details.

Reviewer 2 Report

This paper reviewed and integrated the research status, research hotspots comprehensively, and research trends of residential carbon emissions. I think this paper is an interesting and hot topic. But it needs a major revision before accepted. Some modifications should be conducted as follows:  

1. The introduction describes too much research background and does not highlight the key points.

2. The innovation of this paper is not obvious. Compared with previous studies, the innovation of this paper and its contribution to this field are increased.

3.  Figure 5 and Figure 6 show similar content, with suggestions for optimization.

4. The empirical part mainly expounds the previous analysis and lacks its own analysis.

5.  The first half of the conclusion is lengthy.

6.  The reference format is not standard, such as line 46 and so on.

7.  As this is a time-sensitive review paper, it is suggested to add literatures in 2020 and 2021 to reflect the latest research progress.

8.  The author needs to check the full text and pay attention to the expression, Such as lines 203-213.

Author Response

Response to Reviewer 2 Comments

Point 1: The introduction describes too much research background and does not highlight the key points.

Response 1: Thank you very much for your suggestions. Based on the comments of reviewers, we revised the introduction. In the first and second paragraphs of the introduction, we briefly introduced the generation of greenhouse effect and its impact on climate change and natural environment, reviewed the policies and research reports on carbon dioxide emission made by international organizations, and briefly introduced the responses made by various countries to reduce carbon dioxide emission. We believe that the introduction of these research background is necessary to better introduce the research topic and help readers understand the relevant background knowledge. In the third paragraph of the introduction, we added the contribution of residential carbon emissions to the total carbon emissions in various countries, further emphasizing the importance of research on residential carbon emissions. In paragraphs 3 to 5 of the introduction, we analyzed the research status and deficiencies of residential carbon emissions, emphasized the significance of this research for reducing carbon dioxide emissions, and briefly introduced the content arrangement of the paper. See the “1. Introduction” of the paper for details.

Point 2: The innovation of this paper is not obvious. Compared with previous studies, the innovation of this paper and its contribution to this field are increased.

Response 2: Thank you very much for your suggestions. Discussing how to mitigate the greenhouse effect and achieve the goals of carbon peaking and carbon neutrality has become one of the hottest research topics in the world. Statistics show that more than 70% of the total carbon emissions originate from residential consumption. By reviewing the literature, we found that the current research on residential carbon emissions mainly focuses on empirical analysis of quantitative measurement and its impact mechanism, residents' willingness to low-carbon consumption through mathematical models. At present, there is a lack of review papers on the research trends and hotspots in this research field. Therefore, combined with visual knowledge map technology, we sorted out the general situation and hot trends of the residential carbon emission research field from a macro perspective. This paper can provide a reference for scholars to understand the development of the field of residential carbon emission research and research trends. In the introduction section of the paper, we describe the innovation and significance of the paper. See the “1. Introduction” of the paper for details.

Point 3: Figure 5 and Figure 6 show similar content, with suggestions for optimization.

Response 3: Thank you very much for your suggestions. Table 2 has shown in detail the results of cluster analysis of publication keywords by using CiteSpace software, which is repeated as shown in Figure 6. After careful consideration, we removed Figure 6. See the "4.1 Research Hotpots" of the paper for details.

Point 4: The empirical part mainly expounds the previous analysis and lacks its own analysis.

Response 4: Thank you very much for your suggestions. In the empirical section of section 4, we have revised some content. See the "4.1 Research Hotpots" of the paper for details.

Point 5: The first half of the conclusion is lengthy.

Response 5: Thank you very much for your suggestions. Based on the comments of reviewers, we redivided the conclusions into two sections of conclusions and prospect. In section 5.1, we have made the main conclusions clearer by writing them in points. In section 5.2, we added the description of the research trend of future residents' carbon emissions, and put forward suggestions on the national response to low-carbon development. See the “5. Conclusions and Prospect” of the paper for details.

Point 6: The reference format is not standard, such as line 46 and so on.

Response 6: Thank you very much for your suggestions. We have rechecked and revised the format of the references in the paper. See the “References” of the paper for details.

Point 7: As this is a time-sensitive review paper, it is suggested to add literatures in 2020 and 2021 to reflect the latest research progress.

Response 7: Thank you very much for your suggestions. We researched the papers related to residential carbon emissions in the recent three years, and selected 26 papers as references to be included in the section of introduction and empirical analysis of the papers. See the “1. Introduction” and “4.1 Research Hotspots” of the paper for details.

Point 8: The author needs to check the full text and pay attention to the expression, such as lines 203-213.

Response 8: Thank you very much for your suggestions. We have checked and revised the grammar and format of the whole paper based on the journal standards. See the paper for details.

Reviewer 3 Report

The theme of this paper is clear. The CiteSpace system is used to study the carbon emissions of residents, clarify the change of research focus, and also has innovative significance for suggestions on future research directions. But it is worth noting that "The period was selected from January 1, 2000, to December 31, 2021. In the Web of Science database, we used 'resident CO2 emission', 'resident carbon emission' as the subject words to guide the literature search and got 1215 records." It cannot be verified repeatedly. The authors need to provide some sophistication in terms of methodology elevating the level of innovation in the contribution. In addition, the number of references cited in recent five years is relatively small.

Author Response

Response to Reviewer 3 Comments

Point: The theme of this paper is clear. The CiteSpace system is used to study the carbon emissions of residents, clarify the change of research focus, and also has innovative significance for suggestions on future research directions. But it is worth noting that "The period was selected from January 1, 2000, to December 31, 2021. In the Web of Science database, we used 'resident CO2 emission', 'resident carbon emission' as the subject words to guide the literature search and got 1215 records." It cannot be verified repeatedly. The authors need to provide some sophistication in terms of methodology elevating the level of innovation in the contribution. In addition, the number of references cited in recent five years is relatively small.

Response: Thank you very much for your suggestions. Our responses to your suggestions are as follows:

(1) Because this paper was written in April 2022, the search results at that time were 1215 records. After verification, we found that the search result changed to 1,261 records. The reason may be that more journals have been included in the web of science, so the results of the current search are not consistent with those of the paper. Due to this uncertainty, in order to ensure the accuracy of each data in this paper, we deleted the specific search results after consideration, and revised the content of data sources. See the “2.1. Data Sources” of the paper for details.

(2) We re-searched the papers related to residential carbon emissions in the recent five years, and selected 26 papers as references to be included in the section of introduction and empirical analysis of the papers. See the “1. Introduction” and “4.1 Research Hotspots” of the paper for details.

Round 2

Reviewer 2 Report

The authors have made a comprehensive revision according to the review comments.I have no problem, I suggest receiving it.